# Psychological and Behavioral Impacts of the COVID-19 Pandemic on Children and Adolescents in Turkey

**DOI:** 10.3390/ijerph192316207

**Published:** 2022-12-03

**Authors:** Aysel Esen Çoban, Nilay Kaptan

**Affiliations:** Department of Early Childhood Education, Faculty of Education, Hacettepe University, Ankara 06230, Turkey

**Keywords:** COVID-19, psychological symptoms, behavioral changes, physical activity, children, adolescents

## Abstract

The present study aimed to investigate the physical, psychological, and behavioral alteration in children in the age range between 3 and 18 years before and during the COVID-19 pandemic from the perspective of parental perception. In this study, the survey model was used as a quantitative research method. A snowball sampling method was used, and 841 mothers participated. Descriptive statistics, one-way analysis of variance, related samples t-test, Wilcoxon signed-rank test, and stepwise regression analysis were used to analyze the data. It was found that the physical, psychological, and behavioral negative impacts of the COVID-19 pandemic varied depending on the developmental stages of the children. Compared to the pre-pandemic period, an increase in children’s screen time, as well as a decrease in their physical activity time, was observed during the pandemic. In addition, it was found that screen time, physical activity time, and the square meters of the house are among the significant predictors of mood, behavioral changes, and nutritional problems in children. In terms of anxiety symptoms, physical activity time and screen time were found to be significant predictors. In addition, screen time, age, and physical activity time were observed to be significant predictors of cognitive change symptoms.

## 1. Introduction

The COVID-19 virus, which has economic, sociological, and psychological effects, as well as the many losses of life it causes all over the world, first appeared in the Wuhan province of China [1] and was declared a pandemic by the World Health Organization on 11 March 2020 [2]. With the outbreak of the epidemic, efforts to control it around the world have caused travel bans and restrictions [3]. In addition to socioeconomic effects, such as economic recession, slowdown in the international stock market, starvation of poor people, and unemployment of working people [4,5], these restrictions have also had psychological effects, such as depression, anxiety, stress, and sleep disturbance [6,7,8], in addition to overload on the health system.

Turkey is among the countries that have severely experienced the socioeconomic and psychological effects of the COVID-19 epidemic, with a total of 16,919.638 million cases and 101,203 deaths [2]. Studies conducted on the psychological effects of the epidemic have determined that adults in the country show moderate and high symptoms of hopelessness and anxiety [9], while the level of negative effects on parents is high, and it also results in an increase in the perceived stress level of children [10]. In addition, it has been observed that children in the country have started to exhibit behaviors, including “anxiety, fear, aggression/anger, hyperactivity, sibling jealousy, dependency on parents” after the pandemic [11]. Thus, it can be said that children have been impacted by the pandemic as much as their parents. More importantly, children face the risk of suffering the most as a result of the economic, social, and psychological problems brought about by the epidemic, although they are not seen as direct victims of COVID-19 [12,13].

Turkey is one of the two countries where schools have been closed for the longest period of time, with a total of 47 weeks of closure due to the pandemic, and almost 25 million students who had to attend school in the country received education via distance education. It is known that school reduces children’s calorie intake and screen time by courtesy of physical activity [14], hence helping to protect children from obesity and screen addiction. Moreover, it is known that physical activity is significantly dependent on the type of house, such as square meters, having a balcony, or a garden [15]. Research has shown that as the square meters of the house increase, the chance of children to move freely increases [16,17] since they probably had more space and better circumstances to exercise. In addition, there is a significant relationship between physical activity and screen time [18], and screen time is closely related to the psychological well-being of children and adolescents [19]. It has been found that children’s screen time increased by 87% during the COVID-19 period [20]. 

In addition, some studies investigating the impact of the COVID-19 epidemic process on children of different age groups found that children between 9 and 12 years are more anxious than other age groups, while children aged 13 to 18 years have more sleep disorders compared to other age groups [21]. Moreover, a study conducted in China found that younger children (3–6 years old) showed more symptoms of fear about their family members contracting the COVID disease, and they developed more overattachment [22]. In addition, in a study of 6510 children between the ages of 4–10, more than half of the children showed an increase in irritability, intolerance to rules, caprice, and demands, especially in the 4–6 age group [23]. 

There are international and comparative studies investigating the reflection of the COVID-19 pandemic on children [8,24,25,26], whereas there are few national studies examining the psychological and behavioral effects of the epidemic on children [10,11,27]. Therefore, it is clear that there is a gap in the literature, especially at the national level. We claim that the present study holds importance in terms of investigating the impact of pandemic experiences on the 3–18 age group in order to conduct a comprehensive analysis of the pandemic, considering that schools have been closed for a long period of time in Turkey, and the curfews have always covered individuals who are under 18. 

The United Nations Development Program states that the pandemic has had a debilitating all over the world; therefore, it is the responsibility of each country to respond promptly and effectively [28]. For this reason, every single study conducted during the pandemic period will be a resource of information for program makers, educators, psychologists and psychiatrists, teachers, and parents when developing protective and preventive programs and training after the pandemic. In addition, due to the fact that the present study was conducted at the end of 15 months of pandemic, it will constitute guidance for the evaluation of the chronic psychological and behavioral impact on children aged 3–18 years and for the development of intervention and prevention programs for permanent psychological symptoms that may occur. 

Within this context, the study aims to (a) examine before and during pandemic psychological and behavioral changes psychological and behavioral changes (anxiety, mood, sleep problems, behavioral alteration, feeding, cognitive alteration) that are caused by COVID-19 in children at different developmental stages, (b) analyze the differences between children’s habits (screen time, physical activity) before and during the COVID-19 pandemic, and (c) examine how children’s gender and age, mother’s employment status, father’s employment status, children’s habits, and home conditions predict psychological and behavioral changes of children and adolescents.

## 2. Materials and Methods

### 2.1. Research Model

A survey model was used in this quantitative study. The survey model is defined as a pattern that aims to collect data with the purpose of determining the characteristics of a group, such as opinions, attitudes, abilities, knowledge, and beliefs [29].

### 2.2. Participants

The participants of this study were 841 parents with children aged 3–18 years. Data were obtained from parents by snowball sampling, which is a nonprobability sampling method [30]. In this method, participants in the study refer the study to other people who could potentially participate or contribute through their social networks [31]. This method is compatible with the purposive sampling method [32], and within the scope of the method, the participants of the study are not randomly selected [33]. In this case, this is a limitation because there is a possibility that the sample of the study is not representative of the general population. The researchers selected the first 50 participants in the study from parents with children aged 3–18 from different cities in Turkey. These participants were then asked to deliver the measurement tools of the study to potential participants through their social networks (Facebook, Twitter, Instagram, and WhatsApp). Since the data of the study were collected during the pandemic period, face-to-face data collection was not allowed. For this reason, data were collected online, and the snowball technique was used in order to reach parents with children aged 3–18.

The data were collected from a total of 975 participants, but 134 participants were excluded from the sample of the study in order not to adversely affect the reliability of the study, as they were not able to provide the correct answer to the control question that was used as a measurement tool by the researchers. Participants who were likely to be distracted or unable to focus while filling out the evaluation form using the control question were not included in the study.

Table 1 shows the participants’ demographics. There were 424 female and 417 male children in this study. In terms of developmental period, the number of children in adolescence was 399; in middle childhood, it was 193; and in early childhood, it was 249. When Table 1 is analyzed in terms of the employment status of the mothers, it is seen that 242 mothers were working full time, and 18 mothers had lost their jobs due to COVID-19. It was also among the findings of the study that almost one out of four mothers was unemployed. Similarly, it was seen that 451 fathers were working full-time, and 13 fathers had lost their jobs due to the COVID-19 pandemic. When the characteristics of the houses of the participants were examined, it was determined that 433 houses were between 100–149 m^2^, and 563 houses had balconies.

### 2.3. Data Collection Process

This study aimed to investigate the physical, psychological, and behavioral changes of children aged 3–18 years before and during the COVID-19 pandemic from the perspective of parental perception by using a screening model. The “Psychological Impact of the COVID-19 Pandemic on Children and Adolescents Questionnaire” was created online in Google Forms and was used as a data collection tool. Data were collected online at one time via WhatsApp, Instagram, Facebook, and email. In the study, parents were asked how their children’s behavior and psychology differed during the pandemic, considering the pre-pandemic period. It was thought that while the pandemic was still going on, they could better evaluate their children’s experiences in the process by thinking about their children’s previous behaviors and psychologies. Before completing the survey, information about the objectives of the study was provided to the participants, and informed consent was requested. The participants could not complete the survey without giving permission by clicking on the consent form. Approval from the Ethics Board of the authors’ institution (Hacettepe University) was obtained for the research.

### 2.4. Data Collection Tool

The created data collection tool consisted of three parts. While the first part included the demographic variables of parents and children (see Table 1) in the study, the second part included questions about children’s and adolescents’ screen use and physical activities before and during the pandemic, according to parental perceptions. The final section included the “Impact Scale of the COVID-19 and Home Confinement on Children and Adolescents Survey,” which aims to examine the psychological and behavioral impact of the COVID-19 pandemic on children and adolescents.

The Impact Scale of the COVID-19 and Home Confinement on Children and Adolescents Survey [34] has 31 symptoms on a scale ranging from 1 (much less compared to before quarantine) to 5 (much more compared to before quarantine), and the questionnaire consists of six dimensions as follows: anxiety (e.g., my child is worried), mood (e.g., my child is sad), behavioral alterations (e.g., my child is bored), sleep problems (e.g., my child is having nightmares), feeding (e.g., my child has no appetite), and cognitive alterations (e.g., my child is experiencing difficulty in concentration) [25]. Cronbach’s alpha was calculated as (α = 0.91). For the subdimension Anxiety (α = 0.89), Mood (α = 0.84), Sleep (α = 0.90), Behavioral disturbances (α = 0.85), Feeding (α = 0.27) (which includes only two items that evaluate opposite aspects: No appetite and Eats a lot), and Cognitive disturbances (α = 0.75) were calculated. In this study, Cronbach’s alpha was calculated as (α = 0.94). For the subdimension Anxiety (α = 0.88), Mood (α = 0.82), Sleep (α = 0.73), Behavioral disturbances (α = 0.76), Feeding (α = 0.31), and Cognitive disturbances (α = 0.67) were calculated.

Two items were used to evaluate the screen time in the study. One of them was related to Children’s Screen Usage Time Before Quarantine (item 1), and the other was about Screen Usage Time During Quarantine (item 2). Each item consisted of six answer options. These are “less than 30 min”, “30 to 60 min”, “60 to 90 min”, “Between 90 and 120 min”, “between 120 and 180 min”, and “more than 180 min”.

Two items were used to evaluate the physical activity time in the study. One of them was related to Children’s Physical Activity Time Before Quarantine (item 1), and the other was about Physical Activity Time During Quarantine (item 2). Each item consisted of six answer options. These are “less than 30 min”, “30 to 60 min”, “60 to 90 min”, “between 90 and 120 min”, “between 120 and 180 min”, and “more than 180 min”.

### 2.5. Data Analysis

The IBM SPSS 25 package program was used for the analysis of all the data. Depending on the fulfillment of Assumption of Normality, it was decided whether the analyses would be parametric or non-parametric. In the Assumption of Normality, the skewness coefficient (SC) in the range of ±1.96 means that the data do not show any deviation from normality [35]. In this context, a related sample t-test was used in the analysis of screen time before and during the COVID-19 epidemic. This test is used to determine whether the difference between the two associated sample means differs significantly from each other [35]. The Wilcoxon signed-rank test was used to analyze the physical activity time before and during the COVID-19 pandemic. This test is used when the difference scores do not have a normal distribution [35]. The effect size of the differences, which were found to be statistically significant as a result of the related samples t-test and Wilcoxon signed-rank test analysis, was calculated by the correlation value. In the current study, while calculating the effect coefficients, Cohen’s effect size classification was used [36]. Accordingly, an effect size of 0.20–0.50 is small, 0.50–0.80 is medium, and over 0.80 is considered a large effect. 

Using the one-way ANOVA test, it was determined whether the subdimensions of anxiety, mood, behavioral alterations, sleep problems, feeding, and cognitive alterations of children’s exposure to the COVID-19 pandemic showed a significant difference according to different developmental periods (early childhood, middle childhood, and adolescence). In addition, the Tukey post-hoc test was used to determine the significant difference between the developmental stages of the children. According to Havighurst, infancy and early childhood correspond to the ages 0–5, middle childhood the ages 6–12, and adolescence the ages 13–18 [37]. The age groups were determined accordingly in the present study, and early childhood is considered 3–5 years, middle childhood 6–12 years, and adolescence 3–18 years. In addition, stepwise regression analysis was used in order to test how the independent variables (children’s age, characteristics of the house, screen time, and physical activity time during the COVID-19 pandemic) predict the psychological and behavioral changes of children aged 3–18 years.

## 3. Findings

### 3.1. Psychological and Behavioral Impacts of the COVID-19 Pandemic on Children in Different Developmental Periods

Table 2 shows the findings of the analysis regarding how children are impacted by the COVID-19 epidemic in terms of anxiety, mood, behavior change, sleep problems, feeding, and cognitive alteration subdimensions and of different developmental periods (early childhood, middle childhood, and adolescence). In Table 2, the detailed data show that there are significant differences between children at different developmental stages for almost all items.

In the ANOVA calculation, a significant difference was found between the developmental periods on the basis of total items in the anxiety dimension [F(2, 838) =7.48, *p* < 0.01]. When the anxiety dimension was examined to determine which group was the origin of the difference observed on the basis of the items, it was seen that the averages of children in the adolescence period were higher than the averages of children in early childhood. In this case, it was concluded that adolescent children were more restless [F(2838) = 5.96, *p* < 0.05], worried [F(2838) = 16.49, *p* < 0.001], anxious [F(2838) = 25.60, *p* < 0.01], nervous [F(2838) = 34.29, *p* < 0.001)], afraid of COVID-19 infection [(F(2838) = 14.44, *p* < 0.001], and experiencing physical complaints [F(2838) = 6.50, *p* < 0.05)] than children in early childhood. 

Moreover, there was a significant difference between the total mean scores in the mood dimension according to different developmental stages [(F (2838) = 23.40 *p* < 0.001)]. When the developmental period of these differences was examined, it was found in the mood dimension that adolescent children were more upset [(F(2838) = 9.38, *p* < 0.001)], reluctant [F(2838) = 30.09, *p* < 0.001)], feeling lonely [F(2838) = 11.67, *p* < 0.001)], unsettled [F(2838) = 32.98, *p* < 0.001)], disappointed [F(2838) = 7.87, *p* < 0.001)], and bored [F(2838) = 6.81, *p* < 0.05)] than children in early childhood. When the middle childhood period was examined, it was concluded that the children in the middle childhood period were more upset [F(2838) = 9.38, *p* < 0.001)], reluctant [F(2838) = 30.09, *p* < 0.001)], feeling lonely [F(2838) = 11.67, *p* < 0.001)], unsettled [F(2838) = 32.98, *p* < 0.001)], disappointed [F(2838) = 7.87, *p* < 0.001)], and bored [F(2838) = 6.81, *p* < 0.05)] than the children in the early childhood period.

Considering the total item averages of the sleep problems dimension, it was observed that there were significant differences according to the developmental periods (F (2838) =7.92, *p* < 0.001). An item-based examination of the sleep problems dimension revealed that children in middle childhood and early childhood were more afraid of sleeping alone than children in adolescence. Again, it was concluded that children in early childhood wake up more frequently than children in middle childhood. 

When an examination was made according to the total item scores in the behavioral alteration dimension, it is seen that there are significant differences between the children according to their developmental stages (F (2838) = 6.19, *p* < 0.05). When evaluated on the basis of the items, it was concluded that adolescent children showed more behavioral alterations compared to children in early childhood and middle childhood. According to the feeding dimension, which includes two items, there were no significant differences between the children, depending on their developmental levels. However, an analysis on the basis of items revealed that children in early childhood had less of an appetite than children in middle childhood and adolescence [F (2838) = 8.50 *p* < 0.001], and adolescent children and middle childhood eat more than the children in early childhood [F(2838) = 10.67, *p* < 0.001].

Lastly, in the cognitive alteration dimension, total item score averages were significantly different according to developmental periods (F (2838) = 37.74, *p* < 0.001). On the basis of items, it was determined that adolescent children were more indecisive than middle childhood and early childhood children, and adolescent children experience more concentration problems than early childhood children [F (2838) = 30.88, *p* < 0.001]. In addition, middle childhood children were found to be more indecisive [F (2838) = 28.89, *p* < 0.001] and experienced more difficulty in concentration [F (2838) = 30.88, *p* < 0.001] than early childhood children.

### 3.2. Comparison of Physical Activity and Screen Times of Children Aged 3–18 Years before and during the COVID-19 Pandemic 

A related samples t-test was used to determine whether children’s screen times differed significantly before and during the COVID-19 pandemic. As can be seen in Table 3, there was a significant increase in children’s screen time during the COVID-19 pandemic. While the average screen time of the children was x̄ = 2.64 before the pandemic, it increased to x̄ = 3.98 during the COVID-19 pandemic. The Cohen d value, which was calculated for the effect size, was found to be 1.22. This finding shows that the COVID-19 pandemic process has a significant effect on children’s screen time.

The Wilcoxon signed-rank test were performed to determine whether the physical activity times of children before the COVID-19 pandemic and during the COVID-19 pandemic showed a significant difference, as shown in Table 4. As can be seen, the physical activity time differed significantly (z = 19.2, *p* < 0.001). When the mean rank and totals of the difference scores were taken into account, it was seen that this difference was in favor of the negative rank, that is, in favor of the physical activity time before the pandemic. The effect size value was 19.2/√2. (591 + 74) = 0.76. This result showed that the COVID-19 pandemic highly affected children’s physical activity time. In other words, it was found that there was a significant decrease in the physical activity time of children during the COVID-19 pandemic compared to the pre-pandemic period.

### 3.3. Predicting the Impact of the COVID-19 Pandemic on Children Aged 3–18 Years by Children’s Age, Characteristics of the House, Screen Time, and Physical Activity Time during the COVID-19 Pandemic

The results of stepwise regression analysis to predict the impact of the COVID-19 pandemic on children aged 3–18 are given in Table 5. The analysis was completed in two stages in the anxiety subdimension. In the first stage, the physical activity time variable, which explained the most variance with 5% [Freg (1, 839) = 48.547, *p* < 0.001] in the anxiety variable and the increase this variable created in R2, was significant [Fchange (1, 839) = 48.547, *p* < 0.001]. There was a negative relationship between anxiety and physical activity time. The anxiety score decreased as the physical activity time during the COVID-19 pandemic increased. In the second stage of the analysis, screen time contributing 3% [Freg (2, 838) =48.547, *p* < 0.001] to the variance was included, and thus the explained variance was obtained. The anxiety score increased as screen time increased. The increase that the screen time created in R2 was significant [Fchange (1,839) =28.689, *p* < 0.001].

The analysis of the sleep subdimension was carried out in three stages. In the first stage, the age variable explaining the total variance at the rate of 1.5% was included in the analysis. [Freg (1, 839) = 13.175, *p* < 0.001]. There was a negative relationship between the age variable and the negative sleep symptoms that children exhibited during COVID-19. In addition, the increase that was created by the age variable in R2 was significant [Fchange (1, 839) = 0.015, *p* < 0.001]. It can be said that the negative symptoms related to sleep decreased as the age of the children increased. In the second stage, screen time during the pandemic, which contributed 2% to the variance, was included [Freg (2, 838) = 13.642, *p* < 0.001]. The increase provided by this variable in R2 was significant [Fchange (1, 838) = 0.016, *p* < 0.001]. The total score for the sleep problems subdimension increased as screen time increased. The last variable included in the analysis and contributing 1% to the variance was physical activity [Freg (3, 837) = 12.159, *p* < 0.001]. The increase provided by this variable in R2 was significant [Fchange (1, 837) = 0.010, *p* < 0.001]. Thus, the explained variance rate was 3.5%. There was a negative relationship between the physical activity time variable and the total score of the sleep problem subdimension during the pandemic process. It was seen that the sleep problems experienced by children were decreased as their physical activity increases.

The analysis of the behavioral alteration subdimension consisted of three stages. In the first stage of the analysis, screen time during the pandemic was included in the analysis, which explained the most variance with 5% [Freg (1, 839) = 42.487, *p* < 0.001]. The increase in R2 created by screen time was significant [Fchange (1, 839) = 42.487, *p* < 0.001]. Because there was a positive relationship between the two variables, it can be said that behavioral problems in children increased as screen time increased. In the second stage, physical activity, which contributed 2% to the variance, was included [Freg (2, 838) = 30.825, *p* < 0.001]. The increase provided by this variable in R2 was significant [Fchange (1, 838) = 18.288, *p* < 0.05]. It can be seen that there was a positive relationship between physical activity and behavioral alteration scores. The size of the house (m^2^) variable, which was included in the third place in the analysis, contributed 1% to the variance, thus increasing the total explained variance rate to 8% [Freg (3, 837) = 24.013, *p* < 0.001]. The increase provided by this variable in R2 was significant [Fchange (1, 837) 9.745, *p* < 0.05]. It can be said that as the size of the house increased, negative behavioral alterations decreased.

In the feeding subdimension, the analysis was performed in three stages. Screen time [Freg (1, 839) = 11.041, *p* < 0.01], which explained 1.3% of the variance, was included in the first step, the size of the house explaining the variance at 1% [Freg (2, 838) = 8.864, *p* < 0.001] was included in the second step, and physical activity during the pandemic, contributing 1% to the variance [Freg (3, 837) = 7.535, *p* < 0.001], was included in the third stage of the analysis. The variables in the first [Fchange (1, 839) 11.041, *p* < 0.01], second [Fchange (1, 838) 6.612, *p* < 0.05], and third [Fchange (1, 837) 4.797, *p* < 0.05] rank resulted in a significant increase in R2. Thus, the variance explained for the feeding subdimension was 3%. While there was a positive relationship between the feeding scores and screen time of children and adolescents during the pandemic, there was a negative relationship between the characteristics of the house and physical activity scores during the pandemic.

Table 5 shows that there is a negative relationship in the cognitive alteration subdimension. The analysis consists of three stages. Screen time was the most included in the analysis, with a 9% variance explanation rate in the first stage [Freg (1, 839) = 86.349, *p* < 0.001]. The increase provided by screen time in R2 was significant [Fchange (1, 839) 86,349, *p* < 0.001]. As screen time increased, children’s cognitive alteration symptoms also increased. In the second stage, the age variable, which explained the variance at a rate of 3%, was included [Freg (2, 838) = 57.113, *p* < 0.001]. The increase provided by the age variable in R2 was significant [Fchange (1, 838) 25.370, *p* < 0.001]. There was a positive relationship between the age variable and the cognitive alteration scores of the children. In other words, as the age of children increased, their symptoms of cognitive alteration during COVID-19 decreased. The variable included in the analysis at the last stage and contributing 1% to the variance was the physical activity times of the children during the pandemic [Freg (3, 837) = 42.096, *p* < 0.001]. The increase provided by this variable in R2 was significant [Fchange (1, 837) 10.734, *p* < 0.001]. Table 5 shows that there is a negative relationship between physical activity time and cognitive alteration scores during the pandemic process. In other words, it was seen that the cognitive alteration scores of children and adolescents decreased as their physical activity times increased.

## 4. Conclusions and Discussion

The present study aimed to investigate the physical, psychological, and behavioral changes of children between the ages of 3–18 before and during the COVID-19 pandemic. The obtained results show that the COVID-19 pandemic has psychological and behavioral impacts regarding the developmental stages of children. Compared to pre-pandemic, it is seen that children’s screen time increased and physical activity time decreased during the pandemic. In addition, screen time, physical activity time, and square meters of the house are among the significant predictors of children’s mood, behavior alteration, and nutritional problems. In terms of anxiety symptoms, it is seen that physical activity time contributes to the variance in the first stage, and screen time contributes to the variance in the second stage and has an impact on the anxiety of children. Moreover, screen time, age, and physical activity time were found to be significant predictors of cognitive alteration symptoms. Lastly, age and screen time variables significantly predicted children’s sleep problems. 

The impact of the COVID-19 pandemic on children’s lives is increasing day by day. Consistent with some studies in the literature [21,22,23], the results of this study showed that children are impacted by the COVID-19 pandemic differently depending on their developmental stage. In this study, it was found that children in middle childhood and adolescence showed more anxiety symptoms than children in early childhood. Similarly, the YoungMinds study emphasizes that the majority of young people experience profound anxiety and that 67% of them have a long-term negative effect of the pandemic on their mental health. Psychological problems, especially emotional problems, are quite common among adolescents as a characteristic of development, and these emotional problems are greatly affected by stressful events [38]. The COVID-19 pandemic is also an important source of stress in children’s lives. Therefore, it is seen that children in adolescence are also impacted by the COVID-19 pandemic in addition to their developmental process, and that their anxiety is higher. In addition, Al-Rahamneh et al. measured the behaviors and emotions of middle childhood children before and during the COVID-19 pandemic and found that these children most frequently experienced boredom, irritability, reluctance, and loneliness [39]. Schools being closed, social distance, full lockdown measures or temporary measures taken by governments during the pandemic, and uncertainty about the future or exams are the factors that can cause boredom, irritability, reluctance, and loneliness in children in this period. According to the OECD, when anxiety about these factors is enhanced, children’s daily lives are significantly affected [40]. 

In addition, it was concluded that children in adolescence and the middle childhood period show more mood-related psychological and behavioral symptoms than children in the early childhood period during the pandemic. Hence, mood symptoms may increase due to social media use. There are numerous uncontrollable false news and disaster scenarios on social media that make this process even more difficult. In addition, the low use of social media in early childhood compared to other periods may explain less mood symptoms because young children are not exposed to fake news and disaster scenarios. For example, 38% of children aged 9–12 years in Europe have their own social media accounts, and this rate rises to 77% by the age of 13–16. The literature shows that the amount of social media use during the COVID-19 pandemic is related to variables such as depression [3], risk perception [41], fear of COVID-19, health anxiety [42], and mood changes [43]. 

One of the objectives of this study was to examine before and during pandemic psychological and behavioral changes at different developmental stages. Therefore, children in early childhood experience more sleep-related problems than children in middle childhood, while children in middle childhood experience more sleep-related problems than children in adolescence during the pandemic. In other words, sleep problems decrease as the age of children increases. Children in early childhood are more likely to have misconceptions about causality, risk, and risk change due to their cognitive capacities [44]. For this reason, it seems inevitable that they will experience sleep problems because they cannot evaluate the risk factors in their environment, such as disease and pandemic, more realistically. For example, thinking that a parent who goes out for work or for any other reason will get sick or even die. In a study conducted in China, it was found that younger children show more symptoms of fear that their family members will get COVID-19 and that they develop more overcommitment [22]. A study conducted in Turkey found that such anxiety states, which developed due to the pandemic, are associated with sleep habits [45]. In addition, according to research by Çetin, emotional and behavioral problems increase as the problems experienced by children in middle childhood increase in relation to sleep [46]. Therefore, the negative emotions experienced by children in this period during the COVID-19 pandemic may also be an explanatory factor for their sleep problems. 

Moreover, children in the adolescence period showed significantly more symptoms of behavior change compared to children in early childhood during the pandemic. This difference may stem from multiple reasons. The first reason is that the stressors of the COVID-19 pandemic can trigger behavioral disorders in adolescents [47]. Epidemics and disasters are thought to be associated with symptoms of post-traumatic stress, depression, and anxiety in adolescents [48]. The second reason why adolescents show more behavioral alteration symptoms may be home-related problems brought about by the COVID-19 pandemic process. According to Guessoum et al., home quarantine, especially during the pandemic, was found to be associated with domestic violence [48]. In addition, it was found that adolescents experienced more conflict with their parents during the pandemic process [49]. The third reason why adolescents experience more behavioral alterations may be that they have not been socially self-actualized enough in this process. According to Imran et al., it is necessary that hormonal changes that come with adolescence and also serve to adapt adolescents to social status, peer group, and relationships come together with social dynamics [50]. When these dynamics are partially missing, they are thought to cause behavioral problems in adolescents. 

In addition, the cognitive change symptoms shown by the children were examined in terms of developmental periods. In the study, it was determined that adolescent children showed more symptoms related to cognitive change than children in early childhood and middle childhood, and children in middle childhood compared to children in early childhood. In this context, cognitive change refers to difficulty in concentration and indecisiveness (see Table 2). It has been shown in the literature that there is a significant relationship between decision-making and problem-solving behaviors in adolescents [51,52]. In particular, the uncertainty [40] and the feeling of helplessness [53] brought about by the COVID-19 process may have impacted the decision-making behavior of adolescents to solve the problems they faced during the COVID-19 pandemic. In addition, one of the variables affecting the increase in difficulty in focusing as the age of children increases may be the duration of sleep. Batejat et al. found that the sleep behavior of children in middle childhood was associated with their attentional behavior [54].

Moreover, while children’s screen time increased, their physical activity time decreased during the COVID-19 pandemic. A study conducted in Canada found that children were less active, played outdoors less, were more sedentary, and engaged in more recreational screen-based activities during the COVID-19 pandemic [17]. According to Carroll et al., children’s screen time increased by 87%, and physical activity time decreased by 52% during the pandemic process compared to the pre-pandemic period [20]. Similarly, studies conducted in Italy, Spain [34], and England [55] also highlighted the finding that screen time increased and physical activity time decreased during the COVID-19 pandemic. During the pandemic, children often had to stay at home for longer periods of time due to school closures and mandatory isolation. This resulted in children having limited interaction with their friends and less physical activity time [56]. However, the WHO emphasizes the need for children to sit less and move more in order to grow up healthy. In addition, failure to meet recommended times in physical activity is responsible for more than 5 million deaths worldwide each year in all age groups. Therefore, more research is needed to determine the social, emotional, and physical risks for children in the short and long term caused by the negative conditions caused by the COVID-19 pandemic process and to determine the consequences of this change in daily habits. 

In the present study, it was found that significant predictors of negative mood, nutrition, and behavioral change signs in children were screen time, physical activity, and square footage of the home. In fact, we argue that the common point of all three variables may be movement, that is, physical activity. It is known that as screen time increases, physical activity decreases [17,18], and as the square meters of the house increase, the chance of children to move freely increases [16,17]. The literature emphasizes that increased physical activity prevents obesity behaviors in children [57] and positively affects mental health [58]. Considering these findings, it can be said that children need to be more active in order to reduce the symptoms of negative mood, nutrition, and behavior changes during the COVID-19 pandemic. In addition, physical activity and screen time are significant predictors of anxiety behaviors in children. Anxiety appears to be the most common psychological symptom experienced by children during the COVID-19 pandemic. Many studies investigating the negative psychological effects of COVID-19 emphasize that children show symptoms of anxiety due to the pandemic [3,25,59]. Therefore, it is very important to explain the predictors of anxiety behavior. It is thought to be quite significant that 79% of the news is learned from social media, especially during the pandemic [3], as screen time significantly predicted anxiety in this study. Mertens et al. found in their study that the fear of COVID-19 and anxiety about health are related to receiving information from social media [42]. Several studies have examined the physical activity time of children during the pandemic compared to the pre-pandemic period [20,24,55]. However, only one study has been found that examines the impact of physical activity on children in this process. Alves et al. found that the increased physical activity of children during the COVID-19 pandemic was associated with reduced anxiety symptoms [60]. 

Finally, age and screen time were found to predict children’s sleep problems, while age, screen time, and physical activity were found to predict children’s symptoms of cognitive change. Many parents also complain about the decrease in their children’s physical activity time and the increase in screen time during the COVID-19 pandemic [20]. However, physical activity means more regular sleep and less screen time [14]. There is consistent evidence that the COVID-19 pandemic has a negative impact on children’s mental health [24]. This shows that the measures taken by governments during the pandemic do not sufficiently take the physical, mental, and educational needs of children into account and that governments are on this point open to criticism. However, all countries must respond effectively to the pandemic process, which has inflicted a deep scar on children because an integrated global response is an investment in our future [28].

## 5. Recommendations

In light of these findings, parents should communicate and interact with their children and give them freedom both inside and outside the home. Program makers and educators should create and implement their programs by paying regard to the current pandemic process. In order to improve the psychological well-being and resilience of children, high-quality psychological support services must be provided by psychologists, psychiatrists, and psychological counselors. Bedtime routines should be established by parents for children with sleep problems (2 > 1; 3 > 1, see Table 2). Because routines are external cues that inform children that it is time to sleep, they calm and prepare children for sleep. Children with higher anxiety scores during the COVID-19 process (3 > 1; 2 > 1, see Table 2) should be supported to build coping strategies that will reduce the stressors of the COVID-19 pandemic because failing to deal with the negative effects of COVID-19 in an effective manner will make it difficult for children to cope with stress [59]. For children with more behavioral problems (3 > 1, see Table 2), communication within the family should be strengthened, and the way should be paved for the child to be more social because children in this period need to be socialized in order to be able to interact socially and adapt to their peer group and relationships [50,61,62]. Children who experience cognitive changes (3 > 1; 2 > 1; 3 > 2, see Table 2) should be supported by educators, program makers, and parents with strategies to support their metacognitive skills. Teachers should plan outdoor activities where children can be more physically active, and families should plan leisure time activities, especially outdoors, where children can stay physically active in order to both reduce the time children spend in front of the screen and increase physical activity time. In this quantitative study, the data were obtained and evaluated in light of the information obtained from the measurement tool. It is recommended to conduct qualitative and mixed design studies for detailed and in-depth evaluations of children’s psychological and behavioral symptoms. Snowball sampling, one of the purposive sampling methods, was used in this study. It is recommended that a random sampling method be used to ensure wide representativeness. 

## 6. Limitations

This study has some limitations. The first limitation stems from the fact that face-to-face data collection has become almost impossible due to the pandemic. For this reason, the data of this study were collected online. The second limitation of the study was that the sample was created using the snowball sampling method. In this case, there is a possibility that the sample of the study is not representative of the general population. The last limitation of the study was the measurement tool used within the scope of the study. The psychological and behavioral symptoms of children aged 3–18 were determined by the psychological and behavioral symptoms that are included in the measurement tool; however, the children may also exhibit different symptoms that are not covered by the scale items.

## Figures and Tables

**Table 1 ijerph-19-16207-t001:** Participant Characteristics of the Study.

Variables	Sub-Groups	F	%
Gender of the Child	Female	424	50.4
	Male	417	49.6
Age of the Child	Early childhood (3–5 y/o)	249	29.6
	Middle childhood (6–12 y/o)	193	22.9
	Adolescence (13–18 y/o)	399	47.4
Mother Employment Status	Self-employed	80	9.5
	Works part-time	84	10.0
	Works full-time	242	28.8
	Unemployed	202	24.0
	Lost job due to COVID-19	18	2.1
	Other	215	25.6
Father Employment Status	Self-employed	213	25.3
	Works part-time	62	7.4
	Works full-time	451	53.6
	Unemployed	25	3.0
	Lost job due to COVID-19	13	1.5
	Other	77	9.2
Characteristics of the House	With garden	189	22.5
	With terrace	47	5.6
	With balcony	563	66.9
	It has only windows	36	4.3
	It has more than one exits	6	0.7
Square-meter of the House	Between 50–99 m^2^	130	15.5
	Between 100–149 m^2^	433	51.5
	Between 150–199 m^2^	187	22.2
	200 m^2^ and above	91	10.8

**Table 2 ijerph-19-16207-t002:** Psychological and Behavioral Impacts of the COVID-19 Pandemic on Children at Different Developmental Periods.

	Total (*n* = 841)		Early Childhood (1) (*n* = 205)		Middle Childhood (2) (*n* = 332)		Adolescence (3) (*n* = 304)		ANOVA TEST	
Anxiety	x̄^a^	sd	x̄	sd	x̄	sd	x̄	sd	F	Post hoc
My child is restless	2.22	1.24	1.99	1.09	2.21	1.21	2.37	1.35	5.96 *	3 > 1
My child is anxious	2.35	1.19	1.98	1.06	2.36	1.17	2.59	1.23	16.49 ***	3 > 1; 3 > 2; 2 > 1
My child is worried	2.33	1.28	1.89	1.03	2.27	1.26	2.69	1.36	25.60 ***	3 > 1; 3 > 2; 2 > 1
My child worries when one of us leaves the house	2.09	1.21	2.24	1.25	2.21	1.23	1.86	1.14	8.75 ***	1 > 3; 1 > 2
My child is nervous	2.35	1.35	1.87	1.06	2.23	1.28	2.81	1.45	34.29 ***	3 > 1; 3 > 2; 2 > 1
My child is afraid of COVID-19 infection	2.56	1.33	2.13	1.21	2.73	1.34	2.66	1.34	14.44 ***	2 > 1; 2 > 3 3 > 1
My child is uneasy	2.11	1.21	1.60	0.93	2.09	1.17	2.46	1.30	32.98 ***	3 > 2; 3 > 1; 2 > 1
My child is easily alarmed	2.10	1.21	1.96	1.16	2.20	1.18	2.10	1.26	2.55	-
My child has physical complaints (headache, stomach ache…)	1.56	0.96	1.42	0.80	1.51	0.90	1.72	1.10	6.50 *	3 > 2; 3 > 1
My child asks about death	1.78	1.13	1.71	1.08	1.97	1.18	1.61	1.08	8.61 *	2 > 3; 2 > 1
Anxiety total	2.17	0.02	1.98	0.051	2.21	0.04	2.25	0.04	7.48 **	3 > 1; 2 > 1
Mood										
My child is sad	2.08	1.20	1.80	1.04	2.10	1.19	2.26	1.29	9.38 ***	3 > 1; 2 > 1
My child is reluctant	2.42	1.42	1.82	1.08	2.45	1.38	2.78	1.52	30.09 ***	3 > 1; 2 > 1
My child feels lonely	2.17	1.31	1.82	1.15	2.18	1.31	2.38	1.37	11.67 ***	3 > 1; 3 > 2; 2 > 1
My child cries easily	2.35	1.30	2.64	1.28	2.41	1.27	2.10	1.31	11.26 ***	1 > 2; 1 > 3; 2 > 3
My child feels frustrated	1.91	1.18	1.63	0.99	1.98	1.17	2.03	1.28	7.87 ***	2 > 1; 3 > 1
My child is bored	3.26	1.45	2.94	1.39	3.41	1.42	3.30	1.49	6.81 **	2 > 1; 3 > 1
Mood Total	2.32	0.034	1.93	0.05	2.36	0.98	2.53	1.06	23.40 ***	2 > 1; 3 > 1
Sleep										
My child has nightmares	1.53	0.92	1.47	0.83	1.56	0.97	1.52	0.944	0.60	-
My child sleeps little	2.05	1.17	2.06	1.09	2.02	1.12	2.08	1.27	0.24	-
My child is afraid to sleep alone	2.06	1.34	2.56	1.35	2.31	1.43	1.45	0.94	58.29 ***	2 > 3; 1 > 3
My child has sleeping difficulties	1.98	1.27	2.04	1.28	2.04	1.26	1.87	1.27	1.84	-
My child wakes up frequently	1.65	1.03	1.80	1.02	1.57	0.96	1.65	1.10	3.27 *	1 > 2
Sleep Total	1.85	0.02	1.98	0.05	1.90	0.04	1.71	0.045	7.92 ***	1 > 2; 2 > 3
Behavioral Alterations										
My child argues with the rest of the family	2.44	1.37	2.08	1.17	2.45	1.38	2.66	1.45	10.98 ***	3 > 1; 2 > 1
My child is irritable	2.44	1.40	2.09	1.26	2.46	1.39	2.66	1.45	10.62 ***	2 > 1; 3 > 1
My child is very quiet	1.88	1.06	1.78	0.97	1.77	0.93	2.08	1.22	7.75 ***	3 > 2; 3 > 1
My child has behavioral problems	1.64	1.01	1.48	0.84	1.67	1.03	1.71	1.08	3.39 *	3 > 1
My child is angry	2.62	1.36	2.28	1.23	2.56	1.36	2.91	1.40	14.20 ***	3 > 2; 3 > 1; 2 > 1
My child is very dependent on us	2.95	1.33	3.25	1.30	3.11	1.29	2.56	1.29	21.53 ***	1 > 3; 1 > 2
Behavioral Alterations Total	2.32	0.02	2.16	0.05	2.33	0.04	2.43	0.051	6.19 *	3 > 1
Feeding										
My child has no appetite	1.98	1.19	2.26	1.29	1.95	1.21	1.83	1.07	8.50 ***	1 > 2; 1 > 3
My child eats a lot	2.09	1.26	1.75	1.03	2.15	1.31	2.26	1.30	10.67 ***	3 > 1; 2 > 1
Feeding Total	2.03	0.02	2.00	0.05	2.05	0.04	2.04	0.04	0.21	-
Cognitive Alterations										
My child is very indecisive	2.28	1.25	1.82	1.03	2.23	1.14	2.65	1.39	28.89 ***	3 > 1; 3 > 2; 2 > 1
My child has difficulty concentrating	2.56	1.40	1.92	1.10	2.72	1.41	2.82	1.43	30.88 ***	3 > 1; 2 > 1
Cognitive alterations total	2.42	0.03	1.87	0.06	2.47	0.060	2.73	0.07	37.74 ***	3 > 1; 2 > 1; 3 > 2

* *p* < 0.05, ** *p* < 0.01, *** *p* < 0.001.

**Table 3 ijerph-19-16207-t003:** T-test Results of Screen Times Before and During the COVID-19 Pandemic.

Screen Time	N	x̄	S	Sd	t	P
Before Pandemic	841	2.64	1.02	840	−36.13	0.00 ***
During Pandemic	841	3.98	1.17			

*** *p* < 0.001.

**Table 4 ijerph-19-16207-t004:** Wilcoxon Signed-Rank Test Results of Screen Time and Physical Activity Before and During the COVID-19 Pandemic.

		N	Mean Rank	Sum of Ranks	z	*p*
Screen Time Before Pandemic-during Pandemic	Negative Ranks	12	378	4536.5	21.9 *	0.000 ***
	Positive Ranks	658	334	220,248.5		
	Ties	171				
Physical Activity Time Before Pandemic-during Pandemic	Negative Ranks	591	346	204,778	19.2 **	0.000 ***
	Positive Ranks	74	225	16,667		
	Ties	176				

* Based on negative ranks ** Based on positive ranks. *** *p* < 0.001.

**Table 5 ijerph-19-16207-t005:** Stepwise Regression Analysis Results for Predicting the Impact of the COVID-19 Pandemic on Children aged 3–18.

Sub-Dimension	Model Predictor		B	SH_B_	β	F_Change_	F_Regression_	R^2^	∆R^2^
Anxiety	1	Physical Activity (During Pandemic)	−1.398	0.201	−0.234 **	48.547 ***	48.547 ***	0.055	0.05
		Constant	24.959	0.541					
	2	Physical Activity (During Pandemic)	−1.278	0.199	−0.214 **	28.689 ***	48.547 ***	0.086	0.03
		Screen Time (During Pandemic)	1.232	0.230	0.178 **				
		Constant	19.779	1.104					
Mood	1	Screen Time (During Pandemic)	1.290	0.165	0.261 **	61.238 ***	61.238 ***	0.068	0.068
		Constant	9.057	0.683					
	2	Screen Time (During Pandemic)	1.168	0.162	0.236 **	44.403 ***	54.404 ***	0.115	0.047
		Physical Activity (During Pandemic)	−0.931	0.140	−0.218 **				
		Constant	11.711	0.776					
	3	Screen Time (During Pandemic)	1.213	0.162	0.245 **	5.572 *	38.325 ***	0.121	0.006*
		Physical Activity (During Pandemic)	−0.940	0.139	−0.220 **				
		M^2^	−0.523	0.221	−0.077 *				
		Constant	12.746	0.890					
Sleep	1 2	Age	−0.108	0.030	−0.124 **	0.015 ***	13.175 ***	0.015	0.015
		Constant	10.363	0.331					
		Age	−0.159	0.033	−0.184 **	0.016 ***	13.642 ***	0.032	0.016
		Screen Time (During Pandemic)	0.484	0.130	0.140 **				
		Constant	8.959	0.500					
	3	Age	−0.183	0.033	−0.211 **	0.010***	12.159	0.042	0.010
		Screen Time (During Pandemic)	0.483	0.129	0.140 **				
		Physical Activity (During Pandemic)	−0.313	0.105	−0.105 *				
		Constant	9.934	0.595					
Behavioral Alteration	1	Screen Time (During Pandemic)	0.964	0.148	0.220 **	42.487 **	42.487 ***	0.048	0.048
		Constant	10.134	0.613					
	2	Screen Time (During Pandemic)	0.893	0.147	0.203 **	18.288 *	30.825 ***	0.069	0.020
		Physical Activity (During Pandemic)	−0.45	0.127	−0.143 **				
		Constant	11.686	0.707					
	3	Screen Time (During Pandemic)	0.948	0.148	0.216 **	9.745 *	24.013 ***	0.079	0.011
		Physical Activity (During Pandemic)	−0.555	0.127	−0.146 **				
		M^2^	−0.628	0.201	−0.104 *				
		Constant	12.929	0.809					
Nutrition	1	Screen Time (During Pandemic)	0.157427	0.047	0.114 **	11.041 **	11.041 **	0.013	0.013
		Constant	3.450	0.196					
	2	Screen Time (During Pandemic)	−0.168	0.048	0.125 **	6.612 *	8.864 ***	0.021	0.008
		M^2^	4.037	0.065	−0.089 *				
		Constant	3.774	0.233					
	3	Screen Time (During Pandemic)	0.161	0.048	0.117 *	4.797 *	7.535 ***	0.026	0.008
		M^2^	−0.172	0.065	−0.091 *				
		Physical Activity (During Pandemic)	−0.090	0.041	−0.075 *				
		Constant	4.037	0.262					
Cognitive Alteration	1	Screen Time (During Pandemic)	0.603	0.065	0.305 **	86.349 ***	86.349 ***	0.093	0.093
		Constant	2.447	0.269					
	2	Screen Time (During Pandemic)	0.453	0.071	0.229 **	25.370 ***	57.113 ***	0.120	0.027
		Age	0.089	0.018	0.180 **				
		Constant	2.142	0.272					
	3	Screen Time (During Pandemic)	0.452	0.070	0.229 **	10.734 **	42.096 ***	0.131	0.011
		Age	0.075	0.018	0.151 **				
		Physical Activity (During Pandemic)	−0.187	0.057	−0.109 *				
		Constant	2.723	2.723					

* *p* < 0.05, ** *p* < 0.01, *** *p* < 0.001.

## Data Availability

The data presented in this study are available on request from the corresponding author. The data are not publicly available due to ethical reasons.

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
