# Peer review of "Psychological and Behavioral Impacts of the COVID-19 Pandemic on Children and Adolescents in Turkey"

_ijerph, 2022, doi:10.3390/ijerph192316207_

Round 1
Reviewer 1 Report
Introduction
- The 3rd paragraph of page 2/19 does not strongly support the argument by talking about the long-term effect of the Covid-19 pandemic 69 on children but ending with evidence-based research to present the 74 psychological and behavioral reactions of children during the epidemic process is an urgent need. (this is also not a long-term effect investigation)
Method
- In light of the fact that the quantitative research technique focuses on generating sample results for the population, the authors should elaborate on the justification of the research design as follows:
- The snowball sampling approach is used, which may have an impact on the representativeness and sampling bias. In this section, the researcher needs to state or provide the rationale for using this method. (It is mentioned in the limitation part, but it should also be mentioned in the method section.)
- Why does the author(s) choose to use the term "hidden population" (line 110, page 3/19)?
- Does the elimination of 134 participants have any impact whatsoever on the sample selection process?
- In the section describing the instruments and inquiries, Insufficient information regarding the standard of assessment, and the definition of each variable, for example, screen time, physical activity, and measuring method. How are changes in psychology and behavior measured? Exists a standard instrument? (as shown on line 144)
- It should be clearly explained that the data consists of a one-time gathering. then request retrospective or alternative strategies How might such an approach impact the data's quality or any limitations?
- The lack of information regarding the spatial coverage of the data collection (despite the fact that it is collected online), unless it can be determined that it covers a certain country, region, or sub-area. It will improve the reader's ability to visualize the situation.
- The term "affected" may not be correct and carefully use (at line 183). This is due to the fact that the statistic utilized for analysis did not describe the causal link of the variable, but rather its correlation.
Results
- The writing style in the second and third paragraphs of the results section (lines 190 and 208, as well as the left parts of this section) did not resemble that of an academic publication. The approach utilized to show the parameters may mislead the reader due to the excessive number of parameters presented without explanation. I believe the reader will be able to follow these criteria from table 2.
- The interpretation of the finding on “3.3. Predicting the Effect of the Covid-19 Pandemic on Children Aged 3-18 Years by Children's Age, Characteristics of the House, Screen Time and Physical Activity Time During the Covid-19 Pandemic” The presentation of the study's findings was difficult to comprehend. Because of the that the issues of two and three stages were addressed in each section of the analysis. Without discussing the purpose or meaning of each stage. How arise imported stage variables are given differently to each section? This makes the presenting of results in this section difficult to comprehend and distracts from the study's findings, as it is too complex to explain each stage individually (I apologize if I misunderstood this section).
- Table 5 appears to be lengthy and difficult to read. Also taking up a great deal of space, it may be placed in the appendix. Please also verify the format and numbers displayed in each table, they appear to be improper.
For example
Recommendation
- This study provides no conclusive evidence about the correlation between financial factors and dependent variables. Nonetheless, the authors begin by stating, “In this direction, governments should provide financial assistance, especially to families who have been financially affected by the pandemic” (line 528). It would be preferable if the author(s) suggested or contributed the result of the finding rather than the general situation or concept unrelated to this investigation.

Author Response
Dear reviewer,
Thank you for valuable comments. We made changes you can see it atteched document
Sincerely
Corresponding Author

Reviewer 2 Report
The article does not bring innovation and income is a variable to be considered with more emphasis in the text. The way the sample was designed and the data collected make the results fragile. I do not recommend publishing.
Author Response
Dear reviwer,
Thank you for reviewing our manuscript
Sincerely
Corresponding Author

Reviewer 3 Report
This topic is of great interest right now. There are definite topics that need to be addressed in order to be of great quality.
The writing quality needs to be improved.
The methods need to be improved. There is no description of each of the assessment subscales, reliability and validity of the assessment subscales, and the number of items for each subscale. It's hard to connect the results and what they really mean when I can't tell what you used for the assessment and how credible it is. Then calculate the reliability of the subscales collected for your study to see how well it relates to the scale/subscales from previous literature.
The results are confusing at times because of all of the subscales and different independent variables. Use tables for the analyses instead of paragraphs to explain each of the calculations with significance or not.
The tables need to be better aligned by column - example table 1 - numbers and decimals are not exactly aligned.
Discussion needs to have more flow between topics. Within each paragraph, make sure that what you found is more in discussion of the topic itself instead of the findings only again.
Author Response
Dear reviwer,
Thank you for valuable comments our manuscript. You can see changes at attached documant.
Sincerely
Corresponding Author

Round 2
Reviewer 1 Report
My only last remaining concern pertains to the term "hidden population" that appeared on line 5 of page 3. Typically, this term was applied to a group of the population, such as sex workers, LGBTQ people, drug users, etc. In this situation, children and adolescents should not be considered "hidden population" unless the author can provide a justification.
Author Response
Dear reviewer,
We made the change in manuscript according to your recommendation. Please see the attachment.
Sincerely,
Corresponding Author
